# Assessing Corneal Endothelial Damage Using Terahertz Time-Domain Spectroscopy and Support Vector Machines

**DOI:** 10.3390/s22239071

**Published:** 2022-11-23

**Authors:** Andrew Chen, Zachery B. Harris, Arjun Virk, Azin Abazari, Kulandaiappan Varadaraj, Robert Honkanen, Mohammad Hassan Arbab

**Affiliations:** 1Department of Biomedical Engineering, Stony Brook University, 100 Nicolls Rd, Stony Brook, NY 11794, USA; 2Department of Ophthalmology, Renaissance School of Medicine, 101 Nicolls Rd, Stony Brook, NY 11794, USA; 3Department of Physiology and Biophysics, Renaissance School of Medicine, 101 Nicolls Rd, Stony Brook, NY 11794, USA

**Keywords:** terahertz time domain spectroscopy, corneal endothelium, corneal tissue hydration, intraocular pressure, corneal edema

## Abstract

The endothelial layer of the cornea plays a critical role in regulating its hydration by actively controlling fluid intake in the tissue via transporting the excess fluid out to the aqueous humor. A damaged corneal endothelial layer leads to perturbations in tissue hydration and edema, which can impact corneal transparency and visual acuity. We utilized a non-contact terahertz (THz) scanner designed for imaging spherical targets to discriminate between ex vivo corneal samples with intact and damaged endothelial layers. To create varying grades of corneal edema, the intraocular pressures of the whole porcine eye globe samples (*n* = 19) were increased to either 25, 35 or 45 mmHg for 4 h before returning to normal pressure levels at 15 mmHg for the remaining 4 h. Changes in tissue hydration were assessed by differences in spectral slopes between 0.4 and 0.8 THz. Our results indicate that the THz response of the corneal samples can vary according to the differences in the endothelial cell density, as determined by SEM imaging. We show that this spectroscopic difference is statistically significant and can be used to assess the intactness of the endothelial layer. These results demonstrate that THz can noninvasively assess the corneal endothelium and provide valuable complimentary information for the study and diagnosis of corneal diseases that perturb the tissue hydration.

## 1. Introduction

The cornea is a clear, avascular tissue that along with the lens focuses incoming light onto the retina. The tissue’s transparency is critical for refracting light and is directly related to the corneal tissue water content (CTWC) [1,2]. The structure of the cornea is displayed in Figure 1. The tissue is composed of five layers, each with different contributions in maintaining regular visual function. Of those layers, the stroma comprises the majority of the corneal tissue volume and is formed from layered collagen fiber bundles and proteoglycans with glycosaminoglycan (GAG) chains. Corneal transparency is due to the lattice-like arrangement of the collagen fibrils minimizing light scattering [3]. High corneal water content disrupts the compact spacing due to the hydrated fibrils spreading apart, resulting in corneal edema and blurred vision [4]. In addition, due to the charged GAG chains, the corneal tissue’s natural proclivity is to continuously imbibe fluid from the aqueous humor [3]. To maintain a state of deturgescence, the water content is regulated by the corneal endothelium layer.

The corneal endothelium’s function is two-fold: allowing nutrients to diffuse into the stroma and regulating stromal hydration. Intercellular tight junctions between endothelial cells act as a porous barrier. Aqueous humor passively diffuses through the barrier into the corneal stroma to deliver oxygen and metabolites to the tissue [5,6]. Endothelial cells regulate CTWC by actively transporting excess fluid out of the stroma via a sodium potassium pump [6,7]. The balance between the two forces can be disrupted by corneal diseases such as Fuchs’ dystrophy [8] or posterior polymorphous dystrophy and ICE syndromes [9], by elevated intraocular pressure (IOP) [10,11,12] or by surgical [13] or mechanical trauma [14] damaging the corneal endothelial layer. Figure 1 (left) shows that the intact endothelial layer actively pumps out excess fluid from the stroma layer in the healthy cornea, whereas the pump function is compromised in the cornea with a damaged endothelium (Figure 1, right), resulting in corneal edema. In addition, endothelial cells have little to no regenerative properties [15]. As a result, cells undergo hypertrophy to maintain a confluent endothelial layer after trauma [16,17]. However, the effectiveness of the endothelial layer diminishes as a result of the reduced cell population. Monitoring CTWC fluctuations resulting from an impaired endothelial layer would provide supplementary information for the early diagnosis of corneal diseases.

Currently, there are few ways to noninvasively assess CTWC. A quintessential technique is to estimate tissue hydration from corneal thickness measurements based on the work by Ytteborg et al. [1], where a linear regression between the two was calculated. However, recent in vivo work has shown that a linear relationship may not sufficiently describe the tissue hydration dynamics under abnormal conditions [18]. Alternative techniques are emerging to better monitor corneal hydration changes. Confocal Raman spectroscopy can directly probe CTWC but requires a high fluence, limiting in vivo applications [19]. Brillouin microscopy measures scatter from a low-powered infrared laser to indirectly assess CTWC via changes in corneal tissue biomechanics [20].

Alternatively, terahertz spectroscopy is a noninvasive imaging modality capable of quantifying the tissue water content. Examples of the biomedical applications of the THz technology include the imaging and diagnosis of cancer tissue [21,22] based on variations in the tissue hydration and index of refraction, measuring picosecond protein vibrational dynamics [23] and diagnosing diabetic foot ulcers [24]. Recent advancements in scanning strategies using a f-θ imaging lens allowed the development of a THz handheld scanner capable of fast imaging at a 2kHz trace acquisition rate [25,26,27]. Other approaches include the use of spatial light modulators in combination with neural networks to achieve high-speed hyper spectral THz images [28]. In vivo studies using the same imaging technologies demonstrate that terahertz time domain spectroscopy (THz-TDS) measurements can grade the burn severity in rodents [29] and porcine [30,31,32,33] animal models based on hyperspectral parameters. Further works have studied the effect of electromagnetic scattering on THz imaging in turbid media [34,35,36,37] and polarization change [38] due to samples.

In this paper, we present our results in probing corneal endothelial damage with THz-TDS reflection spectroscopy. We investigated the pressure damage threshold to ex vivo eye globes by maintaining the IOP at various elevated levels and then returning to physiological conditions. Prior work by our group [39,40] demonstrated how terahertz waves can detect hydration changes in ex vivo porcine corneal samples whose IOP was sequentially elevated from 15 to 55 mmHg in steps of 10 mmHG, whereas normal levels of IOP are between 12 and 22 mmHg [41]. In this study, we investigated the IOP threshold at which pressure-derived endothelial damage occurs. We obtained *n* = 19 porcine globes and controlled their IOP by manipulating the height of a cannulated saline reservoir [42]. Various degrees of endothelium damage were created by increasing each sample’s IOP to either 25, 35 or 45 mmHg before returning to the physiological levels. Our results show that the changes in the spectral slopes between 0.4 and 0.8 THz can be correlated with corneal endothelial layer damage in ex vivo porcine samples. In addition, the THz spectral response to elevated IOP was more dependent on corneal endothelial cell density rather than the magnitude of the pressure insult.

## 2. Materials and Methods

Porcine eyes were used due to their similarity to human eyes [43] and availability. Ex vivo samples were obtained from either a commercial vendor (First VisionTech, Inc, Sunnyvale, TX, USA) or were harvested from euthanized animals at the conclusion of other porcine studies at the Stony Brook University’s Division of Laboratory Animal Research (DLAR). As our corneal samples were obtained from an outside vendor or at the conclusion of approved studies, no IACUC approval was needed. A total of *n* = 19 samples were collected for use in this study. The porcine globes were stored in phosphate-buffered saline (PBS) at 4 °C for up to 3 days before usage as per ex vivo corneal tissue storage [44]. As depicted in Figure 2a, the samples were placed at the focus of our noncontact THz corneal scanning system. A saline reservoir was connected to the anterior chambers of the sample via a 30-gauge needle and polyethylene tubing. The intraocular pressure was manipulated by adjusting the height of the saline reservoir. A pressure sensor, Validyne DP15 (Validyne Inc., Los Angeles, CA, USA), was connected via a 3-way stopcock to monitor the intraocular pressure. All samples were affixed to Styrofoam containers with gelatin to limit motion during IOP measurements.

Using our noncontact corneal scanner [40,45], shown in Figure 2a, terahertz time-domain spectroscopy (THz-TDS) images were formed as the IOP in the ex vivo sample was changed. Terahertz pulses were generated and measured with a photoconductive antenna (PCA) emitter and detector from a commercial THz-TDS system, TeraSmart (Menlo System, Inc, Newton, NJ, USA). A 1560 nm femtosecond fiber laser pumps the InGaAs/InAlAs PCA emitter. A mechanical delay line samples the response of a InGaAs/InAlAs PCA detector with 0.0333 ps resolution. The THz beam is collimated by TPX50 lenses (aspheric, 50 mm focal length, 38.1 mm diameter). The same lens is also for focusing the returned beam on the detector PCA. The silicon beam splitter directs the collocated THz beam upon reflection from the sample towards the detector. Similar to the work by Sung et al. [46,47,48], a 90-degree off-axis parabolic mirror (OAPM), EFL = 50.8 mm; diameter = 76.2 mm, was chosen as the focusing optic. The OAPM has the advantageous property that allows for focusing the broadband THz light without suffering from spherical aberrations or chromatic dispersion. The OAPM can focus the light on the surface of a spherical target whose center coincides with the focal point of the OAPM while maintaining normal incidence over the entire field of view. As a result, the collimated THz beam (~16 mm diameter) can be steered across the aperture of the comparatively larger OAPM, which focuses it onto the surface of an aligned spherical target. As shown in Figure 2a, the 3D-printed housing, containing the THz emitter/detector setup, is translated by two linear stage motors across a grid of precalculated motor positions that map to different angular coordinates on the target’s surface. The spherical sample is aligned to the focal point of the OAPM based on variations in the time of arrival of the reflected THz-TDS pulses. A constant time of arrival of the measured THz-TDS pulses at each pixel of the scan would indicate the sample’s center is well-aligned to the OAPM’s focal point. Further details about the THz corneal scanner design, scanning protocols and alignment strategies are described in our earlier publications [45].

Figure 3 shows the experimental protocol, data collection and machine learning steps used in this study. Each sample was acclimated at the physiological pressure level of 15 mmHg for 1 h before the IOP was elevated to either 25, 35 or 45 mmHg for 4 h. Subsequently, the IOP was returned to 15 mmHg for the remaining 4 h of the experiment to determine if changes in THz reflection signals due to elevated IOP were reversible. From the total of *n* = 19 porcine eye globes, seven samples were assigned to the 25 mmHg group and six samples were assigned to the 35 mmHg and 45 mmHg groups, respectively. The THz-TDS scans with a field of view of ±20° (i.e., 9 by 9 equiangular pixels with 5° spacing) in both axes of the imaging plane were performed every 4 min throughout the entire experimental duration of 9 h. A representative of such a high-pass-filtered THz-TDS signal and peak-to-peak time–domain amplitude image measured from a corneal sample are shown in Figure 2b,c, respectively. At the conclusion of the experiment, the corneas were excised, fixed in 2.5% glutaraldehyde and 1.5% formaldehyde and dehydrated with a sequence of ethanol and hexamethyldisilane (HDMS) solutions in preparation for scanning electron microscopy (SEM) imaging. The fixed endothelial layers were coated in gold (Leica EM ACE600) and imaged with the SEM (ZEISS Crossbeam 360) to assess the endothelial cell density (ECD). The cell densities were calculated based on counting the number of cells in a 100 × 100 micron region of interest (ROI) drawn on the SEM images using ImageJ software (version 1.53 t, Wayne Rasband, Bethesda, MA) [49]. Corneal samples with an ECD greater than 3000 cells/mm^2^ were classified intact [50].

The THz-TDS signals in each pixel were deconvolved with the corresponding signal from reference images of a metallic sphere (R = 8.02 mm) with a similar curvature to the cornea. Both sample and reference measurements were obtained using the same spatial and THz acquisition settings for the deconvolution step. Additional signal processing steps also included the use of a high-pass gaussian filter (>0.1 THz) to attenuate low-frequency noise. As it will be shown in Section 3, the 0.4–0.8 THz region displayed the greatest signal contrast between spectra and, as a result, was chosen as the bandwidth of interest.

Using the mean spectral slopes as predictors, the samples were classified into two groups based on whether they possessed a healthy or damaged endothelium. The mean spectral slopes were determined from the last 30 min of the elevated and physiological IOP time periods and denoted as S_Elev_ and S_Phys_, respectively. The status of each sample’s endothelial barrier was determined based on ECD. Classification of the samples was performed using the support vector machine algorithm (scikitLearn package in Python 3.7). A linear function was chosen as the base kernel for classification in the algorithm. Other kernels such as polynomials, RGB and sigmoid kernels were investigated, but all performed similarly. The model-specific parameter was optimized with grid search cross-validation using accuracy to score each new possible parameter. The bootstrapping procedure was applied to account for the low sample size (*n* = 19) and minimize the variability [51]. The results were averaged from a total of 300 iterations of the SVM model with new training and testing datasets formed in each iteration. The datasets were determined from randomized 70/30 splits.

## 3. Results

In this study, the hydration response to elevated IOP in ex vivo porcine ocular globes was assessed using THz-TDS measurements. Each sample underwent 1 h of acclimation at 15 mmHg, followed by 4 h of elevated IOP at either 25, 35 or 45 mmHg and ending with 4 h of physiological IOP at 15 mmHg. Representative THz-TDS signals measured at the center pixel of an eye globe at the conclusion of each of the three pressure periods are shown in Figure 4a. After performing a Fast Fourier-Transform (FFT) operation, the resulting spectra are displayed in Figure 4b. These THz spectra were deconvolved with the corresponding reference measurement using a metallic sphere of similar curvature and size. Based on these spectral observations, the bandwidth of interest was chosen to be 0.4–0.8 THz. The mean spectral slopes within the bandwidth of interest are shown in Figure 4c over the course of the experimental timeframe.

Based on a prior work, we expect the reflectivity to increase due to the elevated IOP causing increased fluid permeation into the corneal tissue [52]. However, as shown in Figure 4c, two representative samples, 1 and 2, were subjected to the same elevated pressure but differed in their THz responses. In sample 1, the spectral slope changes corresponded with periods of IOP elevation and reduction, whereas, in sample 2, the spectral slope remained relatively constant in spite of the IOP changes. This surprising observation led to the hypothesis that the inter-sample variations in their THz spectra are due to the discrepancy in the intactness of their corneal endothelial layers.

Endothelial cell density (ECD) can be used to assess the integrity of the corneal endothelium. This is because, in damaged endothelium, in order to maintain barrier functionality, the damaged cells undergo hypertrophy to conserve cellular confluency [17,53]. Pre-experimental factors such as the age and health of the animals or sample storage duration may have impacted the corneal endothelial layer integrity prior to our study. It is important to note that, although samples were used sequentially, they were randomly assigned to 25, 35 or 45 mmHg to minimize any bias due to the storage times. ECD values were obtained by counting the number of cells within a drawn ROI in SEM images of the fixed corneal samples at the conclusion of the experiments. Figure 5 shows six examples of SEM images of corneal samples from all three experimental groups (i.e., 25, 35 and 45 mmHg), where, in every group, both intact and damaged endothelium layers were observed. The ECD values for each sample were given at the top of each image in Figure 5.

The results shown in Figure 5 reveal a few surprising findings. Several samples had greater than 7000 cells/mm^2^, which is higher than the mean ECD values of 3500–6500 cells/mm^2^ in healthy porcine cornea [54,55]. The reasons for the higher ECD values may include the curvature of the dried corneal samples, which can result in underestimating the surface area in SEM images, or cell shrinkage during the chemical fixing process [56]. Evidence for the effect of sample curvature in underestimation of surface size and, therefore, overestimation of ECD is that the size of endothelial cells in Figure 5 are, on average, about two times smaller than the typical values. The average axial length of a corneal endothelial cell is 20 µm, whereas the mean axial length in our samples is ~10–12 µm [57]. Regardless of the accuracy of the cell density counts, all groups contained samples with intact endothelium and samples with evident endothelial damage at all three levels of elevated pressure. These unexpected observations may suggest that the intactness and quality of the endothelial tight junctions are directly responsible for the measured THz signal contrasts, not the elevated IOP values.

The relationship between corneal endothelial intactness and the measured THz spectral hyperparameters is depicted in Figure 6. As mentioned in Section 2, the S_Elev_ and S_Phys_ are calculated by averaging the central 25 pixels in every scan during the final half-hour of the elevated and physiological pressure periods. The damaged (<3000 cells/mm^2^) samples had larger absolute spectral slope values, in comparison to samples with greater cell densities. Figure 6 suggests a linear correlation may exist between the two predictors, S_Elev_ and S_Phys_. This linear behavior is expected, because the physiological IOP period of the experiments follows the elevated pressure period. Therefore, the S_Phys_ values are causally dependent on the S_Elev_ values, even if the exact IOP was lowered to 15 mmHg. In other words, if the endothelium of a corneal sample is irreversibly damaged during the elevated IOP segment, it will remain so after the IOP is reduced to physiological conditions.

The scatter plot indicates that damaged samples have greater negative spectral slopes than the healthy samples. The SVM algorithm was applied to see how well these samples can be classified based on the THz spectral responses to elevated IOP. High classification accuracies would provide evidence that the THz spectral parameters can be used as biomarkers for endothelial barrier integrity. As mentioned in Section 2, the samples were classified as either damaged or intact based on their ECD values, with 3000 cells/mm^2^ serving as the threshold value. Three SVM models used one of the three spectral features S_Start_ , S_Elev_ and S_Phys_, and a fourth model used a combination of S_Elev_ and S_Phys_ as the predictors. Similar to the other features, S_Start_ is the mean spectral slopes (0.4–0.8 THz) of the scans of the last 30 min of the acclimation pressure step. The averaged results of 300 separate iterations of the model were used. Each iteration had a newly randomized 70/30 training/test dataset split. The resulting mean ROC curves and ROC-AUC values are displayed in Figure 7. Among these models, the best predictor was the model using both S_Elev_ and S_Phys_ as predictors with an ROC area under the curve value of 0.91 ± 0.12. This suggests the combination of these two parameters may prove to be more robust than using single-parameter classification. One key limitation of this analysis is, however, the relatively low sample size. Future studies are planned to include a larger number of ex vivo and in vivo samples to test the hypothesis formed based on these preliminary experiments.

## 4. Discussion

Extensive research has been performed to demonstrate the applicability of terahertz imaging technology in ophthalmology. The cornea is a fitting target for terahertz imaging due to its avascular tissue structure and its accessibility compared to deep tissue carcinomas. In addition, THz waves have been shown to not affect corneal tissue [39]. Other studies have demonstrated the sensitivity of the terahertz dielectric response to water content changes in ex vivo [40,41] and in vivo corneal samples [42,43,44]. In addition, we presented a finite–difference time–domain method for modeling the THz response to tissue hydration changes. By combining the solution to Fick’s second law of diffusion in conjunction with the Bruggeman effective media and stratified tissue models, we simulated the THz spectral changes as corneal phantoms dehydrated [45]. The initial in vivo rabbit studies by Taylor et al. demonstrated how the reflectivity of 100 GHz waves corresponds to corneal hydration and thickness changes [18,46]. Subsequent work detailed THz corneal scanning geometries similar to this work using 90-degree off-axis parabolic mirrors (OAPM) [47] and another geometry using axicon lenses [48] as focusing optics for imaging the spherical corneal surface while maintaining a normal incidence. Recent work has focused on improving the robustness and alignment of these setups [49,50].

This work investigates the tissue hydration changes that occur after elevated IOP in ex vivo corneal samples using terahertz time–domain spectroscopy. Prior work has demonstrated how THz spectral features can be used to monitor tissue hydration changes. Our attempts to identify a specific pressure threshold for corneal endothelial damage were complicated by the inter-sample variations in tissue hydration response to the same levels of IOP elevation. Using SVM classification and SEM imaging, we show how those differences were directly related to the state of the endothelial layer of each sample. As a result, we have demonstrated that THz spectroscopy can be used to noninvasively assess corneal endothelium functionality.

## 5. Conclusion

In this study, we adjusted the IOP in ex vivo porcine samples to determine the relationship between the measured THz spectral parameters and endothelium intactness. Each sample was exposed to elevated IOP levels of 25, 35 or 45 mmHg for 4 h before returning to physiological pressures (15 mmHg). We found that the pressure-driven tissue hydration changes were more dependent on the extent of endothelial damage rather than the magnitude of the pressure insult. Endothelium damage was assessed via cell density calculated from SEM images. We showed that samples can be classified based on endothelial cell density using the mean spectral slopes as predictors in SVM models. Improved classification was found using the spectral parameters, S_Elev_ and S_Phys_, over models with singular spectral parameters. The combination of these two spectral slope parameters achieved an ROC-AUC value of 0.91 ± 0.12. Further work is needed to demonstrate the predictive power of these THz spectral parameters using both larger sample sizes and in vivo models. Endothelial damage in animal models can be created via scraping the corneal Descemet’s membrane and endothelium [58], transformed endothelial cell injection [58] or elevated IOP [52]. Specular microscopy can be used to confirm endothelial cell density without the artifacts seen using SEM imaging. Using in vivo confocal microscopy and pachymetry measurements, in conjunction with THz-TDS measurements can provide further information on the sensitivity and magnitude of the in vivo corneal tissue response to endothelial damage. THz spectroscopy may potentially provide a non-destructive method of assessing endothelial layer functionality, which provides valuable complimentary information in the diagnosis of many corneal diseases that affect the endothelium.

## Figures and Tables

**Figure 1 sensors-22-09071-f001:**
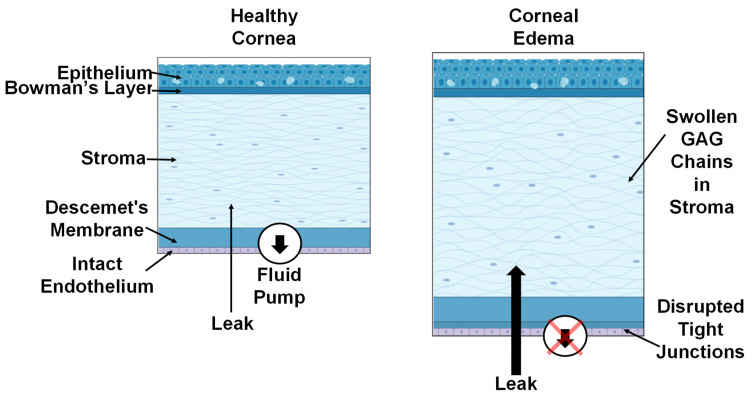
A schematic of two corneas with intact and damaged endothelial layers. The healthy cornea (**left**) is kept within a physiological range of thickness by the endothelial cells actively transporting excess fluid out of the tissue. The edematous cornea (**right**) is a result of the disrupted tight junctions and damaged fluid pump (red X) allowing more liquid into the tissue and inadequate transport of fluid out of the tissue. Images were generated using Biorender.io.

**Figure 2 sensors-22-09071-f002:**
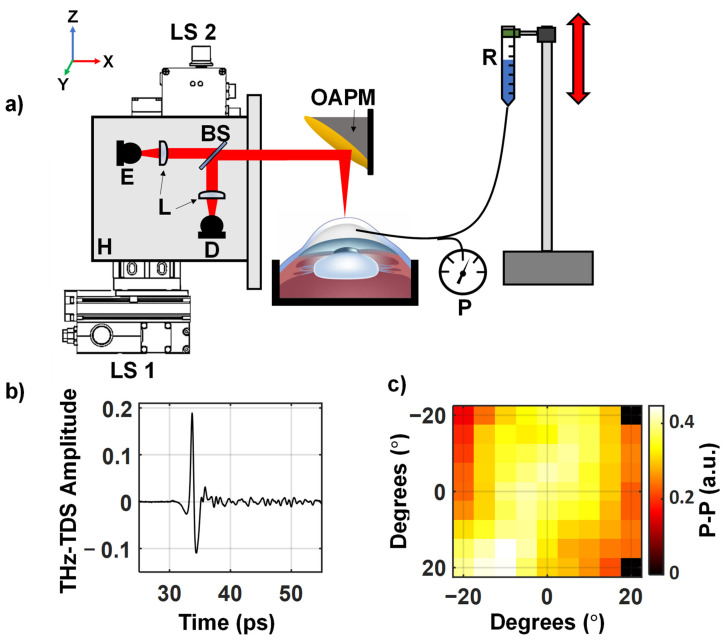
(**a**) A schematic of the setup used in this study. It is similar to one used in a prior ex vivo cornea study [40]. E, emitter; D, detector; BS, silicon beam splitter; R, saline reservoir; L, collimating and focusing lenses (f = 50 mm); P, Pressure sensor; OAPM, off-axis parabolic mirror; H, 3D-printed housing containing the imaging optics, mounted on the LS1 and LS2, linear stage motors, used to move the housing. The red beam shows the path of the THz beam through the setup. (**b**) A representative THz-TDS signal recorded from the center of the corneal sample. (**c**) A raster scan of the corneal sample formed using the peak to peak of the reflected THz pulse.

**Figure 3 sensors-22-09071-f003:**
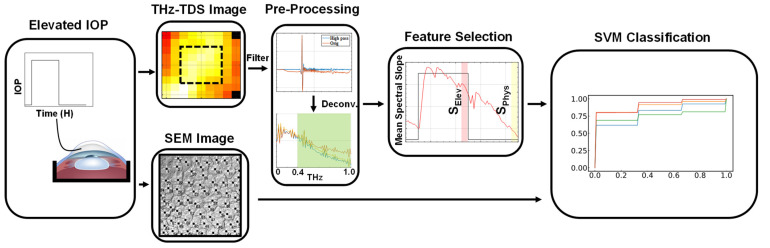
The flowchart depicts the methodology for data collection, signal processing and machine learning classification of the experimental data. The IOP was elevated in porcine ocular globes (*n* = 19) to either 25, 35 or 45 mmHg for 4 h before returning the IOP to the physiological levels (15 mmHg) for another 4 h. During this timeframe, THz-TDS images of the samples were captured. At the conclusion of the experiments, SEM imaging was used to assess the intactness of the endothelium. The THz-TDS signals were high-pass-filtered (>0.1THz) and deconvolved. Spectral slopes were obtained in the bandwidth of interest. The mean spectral slope of the central 25 pixels of the selected ROI was calculated. The mean spectral slope of the last 30 min of the elevated and physiological IOP periods, S_Elev_ and S_Phys_, were used as features for classification using the SVM algorithm.

**Figure 4 sensors-22-09071-f004:**
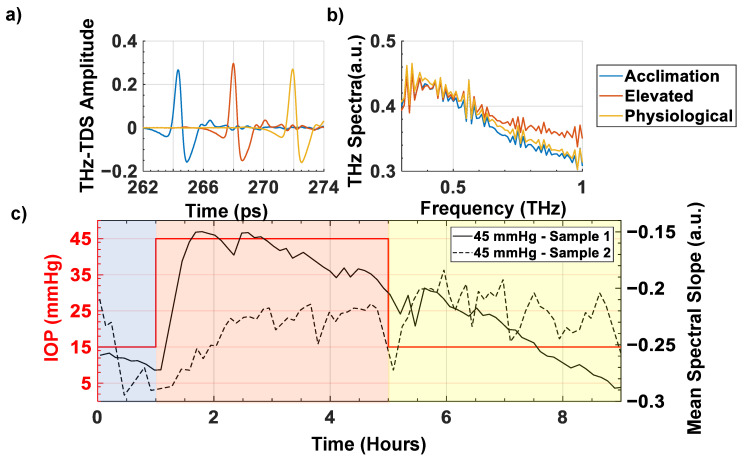
(**a**) The high-pass-filtered THz time–domain signals from Sample #1, at the conclusion of the acclimation period (blue), the elevated period of 45 mmHg (orange) and return to physiological period of 15 mmHg (yellow). (**b**) The deconvolved THz spectra of the same Sample #1. (**c**) The mean spectral slope between 0.4 and 1 THz, averaged over the central 25 pixels, obtained from two different samples, whose IOP were both elevated to 45 mmHg for 4 h. The changes in the intraocular pressure are shown in red color and the left y-axis of the plot.

**Figure 5 sensors-22-09071-f005:**
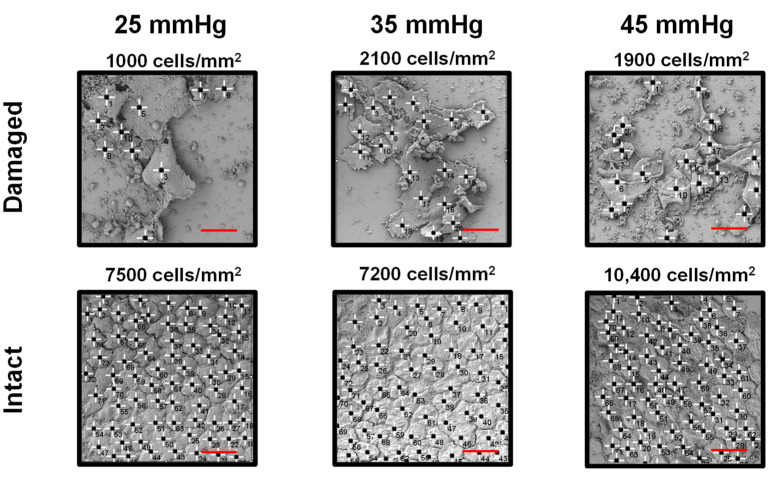
Representative SEM images of damaged and intact endothelial layers from all 3 groups. Each image is 100 by 100 microns. Markers show endothelial cells at the conclusion of the experiment. In the intact group, the endothelial tight junctions are evident. The cell density is estimated based on the number of cells signified by the markers in each ROI and is displayed above its corresponding ROI. The red scale bars are 20 microns in length.

**Figure 6 sensors-22-09071-f006:**
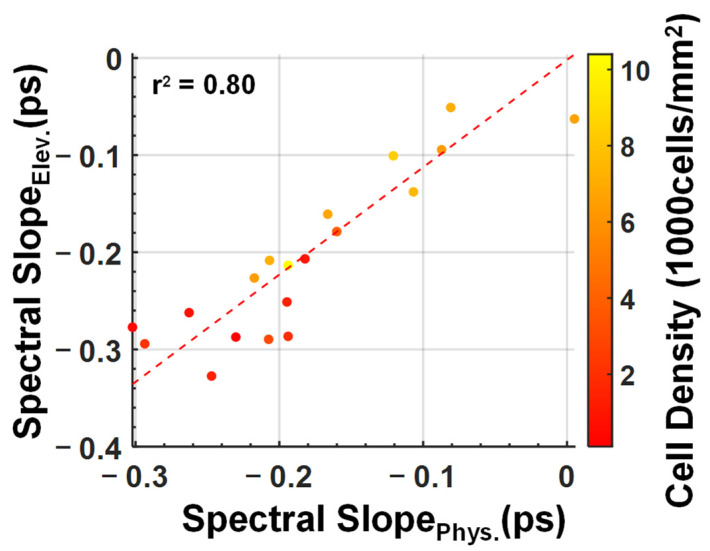
A scatter plot showing the relationship between endothelial cell density and the two spectral slope parameters, S_Elev_ and S_Phys_ in units of picoseconds (ps) The dashed line represents the line of best fit between the two variables. The r^2^ value of 0.80 indicates that there is a strong linear relationship between the two spectral parameters.

**Figure 7 sensors-22-09071-f007:**
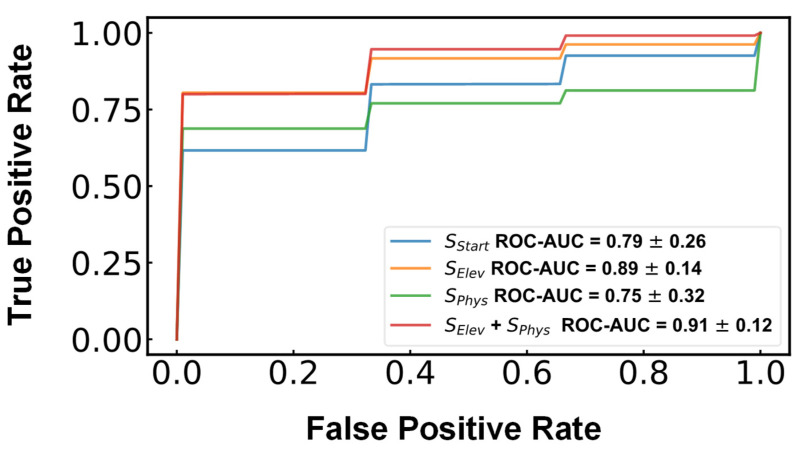
The mean ROC curves of each model are plotted for comparison. The curves and ROC-AUC values are the averaged result of 300 iterations of each model each with a randomized 70:30 training/testing split.

## Data Availability

Data underlying the results presented in this paper are not publicly available at this time but may be obtained from the authors upon reasonable request.

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
