# Peer review of "Assessing Corneal Endothelial Damage Using Terahertz Time-Domain Spectroscopy and Support Vector Machines"

_sensors, 2022, doi:10.3390/s22239071_

Round 1

Reviewer 1 Report

the authors present an interesting study to assess the pressure that determines changes in cornea transparency and to access the corneal endothelial damage using terahertz spectroscopy and support vector machines.

I suggest some changes to improve the overall quality of the manuscript:

- the introduction is definitely too long. I think that the description of the anatomy of the cornea and the function of the endothelium must be reduced or resume

- lines 52-54: the balance between the two forces must be disrupted not only by an elevated IOP but also by the topical therapy used to lower the IOP, as topical carbonic anhydrase inhibitors attenuated the bicarbonate efflux that can lead to corneal decompensation "Abdelghany A, D'Oria F, Alio JL. Surgery of glaucoma in modern corneal graft procedures. Surv Ophthalm. 2021; Mar-Apr;66(2):276-289. doi: 10.1016/j.survophthal.2020.08.002" please introduce this consideration and the reference

-lines 85-100: move to the discussion

-the discussion should start with a resume of the results of the main interest before discussing them properly, thus move lines 108-115 to the first paragraph of the discussion

Reviewer 2 Report

In the current manuscript, the authors induced varying grades of corneal edema by increasing the intraocular pressure, and then assessed corneal endothelial damage with a non-contact terahertz scanner. The study is well-designed, and the manuscript is well-presented. There are a few minor issues that should be addressed.

1. On page 1, lines 29 and 30, the authors should go into more detail about the structure and functions of the corneal layers. (For the reference: https://pubmed.ncbi.nlm.nih.gov/33181005/ and https://www.sciencedirect.com/science/article/pii/S0040816620306285)

2. On page 1, lines 33-35, the authors should discuss the causes of high corneal water content before discussing the damages caused by it.

3. On page 5, lines 180 and 181, the coating process should be mentioned. The coater and SEM device information should be included.

4. On page 5, line 183, the ImageJ software version should be specified.

5. All "et al.," "in vivo," and "ex vivo" should be written in italics throughout the manuscript.

6. The written "Discussion" section is more like a "Conclusion" section. In the "Discussion" section, the current study's findings should be compared to those of other studies. It is suggested that the authors include a "Results and Discussion" section, followed by a clear "conclusions" section.

Reviewer 3 Report

The paper "Assessing Corneal Endothelial Damage using Terahertz Spectroscopy and Support Vector Machines" by Chen et al reports on an interesting technique to assess endothelium damages in an animal model.

I have some comments pertaining to this manuscript.

1)    Material and Methods page 3 lines 117-122. The Authors stated that the eyes were obtained from either a commercial vendor or were harvested from euthanized animals at the conclusion of other porcine studies. From the 19 samples collected for use in this study, how many were obtained from the commercial vendor, and how many were harvested from euthanized animals? Were all these eyes in a comparable state, as we could imagine the handling and shipping of samples coming from a commercial vendor might be different from the animals coming from the same location at the Stony Brook University? I just wanted to clarify whether these two different sources of samples would not create some difference and some sort of heterogeneity in the biomechanical behaviour due to possible post-mortem tissue alterations that might be different from these two providers. 

2)    Material and Methods page 5 line 172. For the sake of clarity and precision, I would suggest minor changes to this sentence: "…six samples were assigned to the 35 mmHg and 45 mmHg groups." I would propose the following: "…six samples were assigned to each 35 mmHg and 45 mmHg groups, respectively." Indeed, a total of 19 eyes were used, and 7 were assigned to the 25 mmHg group, leaving 12 remaining eyes for the 2 other groups. The reader presumes that these 12 eyes were evenly split into two groups of 6 eyes each.

3)    Material and Methods page 5 line 182. For the sake of precision, I would explain in plain words the abbreviation ROI in this sentence: "…100 x 100 micron Region of Interest (ROI)…"

4)    Results page 8 lines 276-278. The Authors found that all groups contained samples with intact endothelium and samples with evident endothelial damage at all three levels of elevated pressure. I would infer from that finding that the group with the lowest IOP, i.e. 25 mmHg, contained samples that might have post-mortem endothelial damage, for the 25 mmHg was not supposed to induce a significant degree of experimental endothelial damage. That leads to my first comment about the possible post-mortem tissue alterations mentioned above.

5)    Results page 9 lines 314-315. The Authors project future studies including a larger number of ex vivo and in vivo samples. I would suggest including, in the in vivo samples, measuring the ECD using current clinical devices. That way, the ECD would be obtained without the problems encountered using the SEM technique (curvature of dried cornea or cell shrinkage)
